# Influence of Preparation Methods on the Concentrations of Lead and Cadmium in *Amanita muscaria* Products

**DOI:** 10.3390/toxics13090734

**Published:** 2025-08-30

**Authors:** Michal Ordak, Wiktoria Gorecka, Pawel Konieczynski, Marek Wesolowski, Alina Plenis, Nasierowski Tadeusz, Bujalska-Zadrozny Magdalena

**Affiliations:** 1Department of Pharmacotherapy and Pharmaceutical Care, Faculty of Pharmacy, Medical University of Warsaw, Banacha 1 Str., 02-097 Warsaw, Poland; s083172@student.wum.edu.pl (W.G.); magdalena.bujalska@wum.edu.pl (B.-Z.M.); 2Department of Analytical Chemistry, Faculty of Pharmacy, Medical University of Gdansk, Gen. J. Hallera 107, 80-416 Gdansk, Poland; pawel.konieczynski@gumed.edu.pl (P.K.); marwes@gumed.edu.pl (M.W.); alina.plenis@gumed.edu.pl (A.P.); 3Department of Psychiatry, Faculty of Pharmacy, Medical University of Warsaw, Nowowiejska 27 Str., 00-665 Warsaw, Poland; tadeusz.nasierowski@wum.edu.pl

**Keywords:** *Amanita muscaria*, lead, cadmium, atomic absorption spectrometry (AAS) analysis

## Abstract

In recent years, the consumption of *Amanita muscaria* has gained popularity as a perceived natural alternative to psychoactive substances, often promoted online for its supposed therapeutic properties. However, the safety of such preparations remains largely unverified, particularly with regard to toxic element content. The present study focused on evaluating how different preparation methods affect the concentrations of cadmium (Cd) and lead (Pb) in the most commonly consumed *Amanita muscaria* products. A total of 50 mushroom caps were collected in October 2024 from the Stąporków Forest District in central Poland. Seven types of preparations were analyzed: dried with skin, dried without skin, tincture (mineralized and direct), boiled, fermented, infusion, and decoction. Samples were analyzed using atomic absorption spectrometry (AAS); depending on the preparation type, some underwent acid digestion prior to measurement. Statistically significant differences in lead concentrations were observed between the various preparation types (*p* = 0.002; η^2^ = 0.97). Lead levels in the direct tincture were significantly higher than those in the boiled (*p* = 0.001), infusion (*p* = 0.001), decoction (*p* = 0.009), and dried without skin (*p* = 0.04) forms. Significant differences in cadmium concentrations were also found (*p* = 0.006; η^2^ = 0.94), with lower Cd levels in the decoction compared to the dried with skin (*p* = 0.009) and without skin (*p* = 0.001), as well as in the fermented form compared to the dried samples (*p* = 0.03 and *p* = 0.004, respectively). These findings demonstrate that the type of preparation significantly influences the levels of cadmium and lead in *Amanita muscaria* products. While aqueous and fermented preparations showed reduced metal concentrations, none can be considered entirely safe, as even lower levels of Cd and Pb may pose health risks with frequent or excessive consumption.

## 1. Introduction

A new psychoactive substance (NPS) is a compound of natural or synthetic origin that affects the central nervous system but is not classified as a psychotropic substance or a narcotic drug. NPSs may pose a health risk or lead to social problems of a similar nature to those caused by psychotropic substances. Globalization and technological progress have significantly accelerated the production and consumption of NPSs worldwide. Easier access to raw materials, the development of the online market, and the rapid spread of information have made new psychoactive substances widely available [1]. By the end of 2021, the European Monitoring Centre for Drugs and Drug Addiction was monitoring approximately 880 new psychoactive substances, including 52 that were registered for the first time [2]. The introduction of additional regulations concerning drug policy has contributed to a noticeable increase in interest in legal and easily accessible substances with psychoactive effects [3]. An example of such a compound is ephedrone, which can be easily produced under home conditions [4]. As restrictions on synthetic psychoactive substances and limits on the sale of precursors have increased, people have begun to seek new alternatives, including those of natural origin. As a result, there has been a noticeable rise in interest in the fly agaric mushroom, which has been used for centuries for ritualistic and medicinal purposes. In response to the growing trend of fly agaric consumption, numerous social media profiles have emerged dedicated to sharing its properties. These platforms present various benefits related to the medicinal effects of *Amanita muscaria*, as well as preparation methods that, according to the authors, make the mushroom safe for consumption [3,5]. Unfortunately, as of now, no scientific studies have confirmed the safety of using fly agaric. A 2023 article published in *Toxics*, based on an analysis of 5600 social media comments, showed that *Amanita muscaria* was most commonly consumed to reduce stress, alleviate symptoms of depression, insomnia, pain, and skin problems. The most frequently chosen form of consumption was tincture among women and dried mushroom among men, indicating a gender-based difference in approaches to its use [3]. Due to this species’ ability to accumulate toxic elements such as cadmium (Cd) and lead (Pb), consumption of this mushroom may pose a risk of heavy metal poisoning [6]. Previous studies published in the scientific literature have examined cadmium and lead concentrations in different parts of *Amanita muscaria*, as well as across various mushroom collection sites. In addition, analyses were conducted to assess differences in element concentrations depending on the harvest period, with studies carried out within the same location [7,8,9]. The most recent study, published in *Toxics* in January 2025 [10], found that the levels of cadmium (Cd) and lead (Pb) in *Amanita muscaria* collected in the Biała Forest in Poland significantly increased with each subsequent month of harvest, with the highest concentrations recorded in October 2023. Average Cd levels in all samples significantly exceeded the permissible limits for dietary supplements: fourfold in August, sixfold in September, and ninefold in October. Although Pb concentrations in the October samples approached the threshold value, they did not exceed it in a statistically significant way. Nevertheless, combined with the high Cd content, this indicates a serious toxicity risk associated with the consumption of this mushroom [10]. The consumption of wild mushrooms, including *Amanita muscaria*, may pose health risks due to their ability to accumulate toxic trace elements such as Cd and Pb. However, while these findings provide important insights into the accumulation of toxic elements in fruiting bodies, they do not answer the question of how much Cd and Pb may remain in processed products actually consumed by users. Therefore, the central focus of this study was to investigate how preparation methods influence the concentrations of Cd and Pb in commonly used Amanita muscaria products. Since tinctures, decoctions, teas, and fermented preparations are currently the most widespread forms of Amanita muscaria use, it becomes crucial to investigate whether these processing methods reduce or retain the toxic metal content. This rationale determined the focus of the present study specifically on Cd and Pb, as two elements most consistently detected at elevated levels in this species. The most common forms of *Amanita muscaria* consumption include boiled, dried, and fermented preparations, as well as decoctions and tinctures. Among five types (boiled, dried, fresh, tea, and tincture), an online survey showed that respondents most frequently consumed the dried form (38%), followed by tincture (22%) and tea (19%) [11]. However, to date, no studies have been reported in the scientific literature on the concentrations of Pb and Cd in specific *Amanita muscaria* preparations. One argument supporting the need for such research is the fact that advocates of *Amanita muscaria* consumption regularly share recipes for various mushroom preparations, which, according to their claims, are intended to reduce the concentrations of toxic elements, including Pb and Cd. They argue that no such studies have yet been conducted, an assertion that is, in fact, true. For this reason, the aim of the present study was not only to determine the concentrations of Pb and Cd but also to evaluate the effect of different preparation methods on their levels in the most commonly consumed forms of *Amanita muscaria*, namely, dried with skin and without skin, tincture, boiled, fermented, infusion, and decoction.

## 2. Materials and Methods

### 2.1. Collection Site

All *Amanita muscaria* samples were collected in the Stąporków Forest District, in the village of Grzybów (Staporkow municipality, Konskie County, Swiętokrzyskie Voivodeship). A total of 50 fruiting bodies of *Amanita muscaria* were collected during a single harvest conducted on 13 October 2024. The mushrooms were collected by persons experienced in field identification of fungi, who had previously published scientific articles on this subject. The collected material was subsequently identified and verified by the authors of this manuscript. The storage conditions were adjusted to the type of preparation: dried mushrooms were stored in paper bags in a dry and shaded place; boiled mushrooms were first kept refrigerated and then frozen until analysis; fermented mushrooms and tinctures were stored in a shaded place; infusions and decoctions were prepared from dried mushrooms and, once made, were kept refrigerated. To enhance the clarity of the analysis and enable comparisons between processing techniques, the preparations were subsequently grouped based on the type of processing used: aqueous (boiled, decoction, infusion), alcoholic (tincture), acidifying (fermented), and dried (with and without skin). The mineralized tincture was not included in the grouping, as it represents a distinct type of preparation due to the use of mineralization, which significantly alters the matrix characteristics.

### 2.2. Preparation of Amanita muscaria Samples

#### 2.2.1. Dried *Amanita muscaria* With and Without Skin

For the dried sample with skin, the stems of the mushrooms were removed and discarded, while the caps were cleaned with a brush to remove debris and then rinsed with water. In this study, only caps were analyzed, as they are most often used as the basis for comparative analyses of toxic elements in mushrooms reported in the literature, and they allow for greater consistency between samples [10]. The mushroom pieces were then evenly arranged on a drying tray and dried in a mushroom dehydrator at 70 °C for 12 h—the initially planned 8 h proved to be insufficient. In the case of the dried sample without skin, the same initial steps were followed; however, the red skin and the yellowish upper layer of the mushroom were additionally removed. The second aim was to compare the concentrations of Pb and Cd in samples of Amanita muscaria with and without cap skin, as, based on studies on other mushrooms, its presence may influence the concentrations of toxic metals [12]. The caps were then cut into long strips and also dried for 12 h at 70 °C [13,14].

#### 2.2.2. *Amanita muscaria* Tincture

To prepare the tincture, 25 g of powdered mushroom caps and 235 mL of 50% ethanol were used. The stems of the mushrooms were removed, and the caps were cleaned and rinsed, then dried at 70 °C for 12 h. The dried caps were then ground in a mill, the appropriate amount of powder was weighed, and mixed with ethanol in a glass jar. The extract was left to stand for 21 days in a dark, dry place, after which it was filtered using a PVDF filter with a pore size of 0.45 μm [15].

#### 2.2.3. Boiled *Amanita muscaria*

For the boiled preparation, 8 g of dried *Amanita muscaria* and 67.4 mL of water were used. The stems were removed, and the caps were cleaned, rinsed, peeled (removing the red skin and yellowish layer), then cut into long strips and dried in a dehydrator for 12 h. The measured amount of dried mushroom was placed in boiling water and cooked for 15 min, after which the mushrooms were strained and thoroughly rinsed [13].

#### 2.2.4. Fermented *Amanita muscaria*

To prepare the fermented form, 98 g of boiled *Amanita muscaria* and 24.5 g of kosher salt were used. The mushrooms were first subjected to a boiling process: the stems were removed, the caps were cleaned, rinsed, peeled (removing the red skin and yellowish layer), sliced into strips, and boiled for 15 min. The boiled mushrooms were then strained and rinsed. Afterward, the ingredients were weighed, the mushrooms were mixed with the salt, and the mixture was placed in a glass jar [13]. The jar was tightly sealed and stored at room temperature in a shaded place. No additional microbial species were introduced, and the fermentation process was spontaneous, occurring under conditions analogous to vegetable fermentation.

#### 2.2.5. *Amanita muscaria* Infusion

To prepare the infusion, 1–3 g of dried *Amanita muscaria* and one cup (250 mL) of distilled water were used. The stems were removed, and the caps were cleaned and rinsed. The mushroom pieces were arranged on a tray and dried for 12 h at 70 °C. The dried mushrooms were then crumbled, placed in a tea infuser, covered with boiling water, and left to steep for 10 min [16].

#### 2.2.6. *Amanita muscaria* Decoction

The decoction was prepared using the same ratio of ingredients as the infusion, namely 1–3 g of dried *Amanita muscaria* and 250 mL of distilled water. The mushrooms underwent the same drying steps: stems were removed, caps were cleaned and rinsed, arranged on a tray, and dried for 12 h at 70 °C. After weighing the ingredients, the crumbled mushrooms were added to boiling water and simmered for 15 min. The decoction was then strained and left to stand for 5–10 min before consumption [16].

### 2.3. Sample Preparation for Analysis

Before proceeding with the determinations, most of the analyzed products were subjected to a mineralization process. To a sample weighing approximately 1.0 g, 20 mL of concentrated nitric acid (V) (HNO_3_) and 5 mL of a 30% hydrogen peroxide (H_2_O_2_) solution were added. The nitric acid (65%, *v*/*v*) was of analytical grade (POCH Basic, Poland), and the hydrogen peroxide (30%, *v*/*v*) solution was pure for analysis (Chempur, Poland). Ultra-pure water was obtained from a Hydrolab UV system (Poland). After 24 h, the samples were filtered through medium-grade qualitative filter paper (PureLand, Chemland, Poland), and then quantitatively transferred to 50 mL volumetric flasks and filled up to the mark with redistilled water. For AAS measurements, gases of spectral purity dedicated to atomic absorption spectrometry were applied. The final solutions were analyzed using atomic absorption spectrometry (AAS), and the results were expressed in micrograms per milliliter (µg/mL) of liquid extract. Since all samples, regardless of their original physical form, were ultimately converted into liquid matrices prior to measurement, reporting results in µg/mL ensured a consistent and methodologically valid basis for comparison across different preparation types. This unit was chosen to reflect the form in which the analytes were quantified and to maintain consistency in interpreting exposure levels for both solid- and liquid-based preparations. Although the original matrices varied, the analytical endpoint was standardized, allowing for the direct comparison of elemental concentrations between all tested forms. In the case of the infusion and decoction, mineralization was not necessary before performing spectrophotometric measurements, as these aqueous preparations were clear solutions without solid residues or alcohol, allowing direct analysis. However, the tincture was analyzed in two variants: after mineralization and directly, in order to verify whether the mineralization step affects the determination of Cd and Pb in this type of matrix.

### 2.4. Determination of Pb and Cd

Samples of *Amanita muscaria* preparations were analyzed using an atomic absorption spectrometer (AAS) SpectrAA 250 Plus (Varian, Melbourne, Australia). Calibration curves for the determination of cadmium and lead concentrations were developed based on the analysis of standard solutions. For cadmium, standard solutions with concentrations of 0.2, 0.4, 0.6, 0.8, and 1.2 μg/mL were used. For lead, standard solutions with concentrations of 0.4, 0.5, 1.0, and 1.2 μg/mL were applied. Based on the obtained data, calibration plots were created with the following equations:A_Cd_ = 0.15182 · C_Cd_ + 0.00523 (r = 0.9986);A_Pb_ = 0.83636 · C_Pb_ − 0.1109 (r = 0.986).

To obtain reliable results, each sample of the *Amanita muscaria* preparations was measured in triplicate. Analyses were carried out using flame-AAS (FAAS). For cadmium determination the lamp current was 5 mA, the slit width was 0.2 nm, and the analytical wavelength was 228.8 nm. For lead, the lamp current was 5 mA, the slit width was 1.0 nm, and the analytical wavelength was 217.0 nm. In both cases acetylene/air was used as the stoichiometric gas mixture and background correction since the deuterium lamp was not applied. No certified reference matrix specific to mushroom preparations was available. Validation was based on calibration with standard solutions and replicate measurements to confirm accuracy and reproducibility.

### 2.5. Statistical Analysis

Statistical analysis was performed using IBM SPSS Statistics 25 (producent). To determine whether there were statistically significant differences between the various *Amanita muscaria* preparations, the Kruskal–Wallis test was applied. To assess the effect size, the eta^2^ measure was calculated, while pairwise group differences were identified using Dunn’s post hoc test. The eta^2^ measure was used to determine the extent to which the type of preparation explains the variation in the concentration of the analyzed element. The statistical analysis included descriptive statistics in the form of minimum and maximum values. A *p*-value of <0.05 was considered statistically significant.

## 3. Results

### 3.1. Type of Amanita muscaria Preparation and Cd and Pb Concentration

In this study, statistically significant differences in cadmium (Cd) concentration were observed between the various Amanita muscaria preparations (H = 16.29; *p* = 0.006; η^2^ = 0.94). The eta-squared value of 0.94 indicates that the type of preparation accounts for 94% of the variability in Cd concentration ranks in the *Amanita muscaria* samples. This reflects a very high variation in cadmium content between the different forms of processing. The highest Cd concentrations were found in dried samples, ranging from 0.1789 to 0.1816 μg/mL (with skin) and 0.1848–0.1888 μg/mL (without skin), while much lower levels were observed in the boiled (0.0294–0.0314 μg/mL), fermented (0.0084–0.0097 μg/mL), decoction (0.0077–0.0090 μg/mL), and tincture (0.0835–0.0854 μg/mL) preparations. In this analysis, the mineralized tincture and infusion preparations were excluded, as in both cases the cadmium concentration was below the detection limit of the analytical method used. A dot plot illustrating Cd concentration in the various *Amanita muscaria* preparations is shown below (Figure 1). Pairwise comparisons using Dunn’s post hoc test revealed that Cd concentration in the decoction was lower than those in the dried with skin (*p* = 0.009) and dried without skin (*p* = 0.001) preparations; in the boiled preparation, it was lower than in the dried without skin (*p* = 0.004); and in the fermented preparation, it was lower than in the dried with skin (*p* = 0.03) and dried without skin (*p* = 0.004) preparations.

Statistically significant differences were also observed in lead (Pb) concentration (H = 22.44; *p* = 0.002; η^2^ = 0.97). The eta-squared value of 0.97 indicates that 97% of the variance in Pb concentration ranks can be attributed to differences among the seven *Amanita muscaria* preparations. This means that the type of preparation had a very strong influence on the content of this element. Pb concentrations ranged from 0.4805 to 0.4841 μg/mL in dried with skin and 0.4447 to 0.4495 μg/mL in dried without skin samples, and 0.3956 to 0.4136 μg/mL in boiled, 0.4566 to 0.4590 μg/mL in fermented, 0.4093 to 0.4106 μg/mL in infusion, 0.4207 to 0.4261 μg/mL in decoction, 0.7675 to 0.7711 μg/mL in mineralized tincture, and 0.7914 to 0.7961 μg/mL in direct tincture samples. Due to the small sample size in each group, the results are presented in the form of a dot plot (Figure 2). Each point (circle) on the plot represents an individual measurement of the analyzed element concentration. To determine which specific preparations differed significantly from one another, Dunn’s post hoc test was applied. The analysis revealed that Pb concentration in the boiled preparation was lower compared to that in the dried with skin (*p* = 0.02), mineralized tincture (*p* = 0.003), and direct tincture (*p* = 0.001) preparations; in the infusion, it was lower than that in the dried with skin (*p* = 0.02), mineralized tincture (*p* = 0.006), and direct tincture (*p* = 0.001) preparations; in the decoction, it was lower than that in the mineralized tincture (*p* = 0.04) and direct tincture (*p* = 0.009) preparations; and in the dried without skin, it was lower than in the direct tincture (*p* = 0.04) preparation.

### 3.2. Amanita muscaria Processing Techniques and Pb and Cd Concentration

The obtained results confirmed the presence of statistically significant differences between the analyzed preparations in terms of cadmium (Cd) concentration (H = 14.31; *p* = 0.003; η^2^ = 0.81) (Figure 3). The eta-squared value for cadmium, similar to that for lead, was 0.81, indicating that the type of processing applied explains 81% of the variation in Cd concentration ranks. Cd concentrations ranged from 0.1789 to 0.1888 μg/mL in dried preparations, 0.0077 to 0.0314 μg/mL in water-based preparations, 0.0084 to 0.0097 μg/mL in fermented preparations, and 0.0835 to 0.0854 μg/mL in tincture preparations. Pairwise comparisons showed that Cd concentration in the dried preparations was higher compared to that in the fermented (*p* = 0.003) and aqueous (*p* = 0.001) forms. Thus, drying resulted in the highest Cd accumulation, whereas aqueous and fermented processing markedly reduced Cd levels.

Statistically significant differences were also found in lead (Pb) concentration (H = 16.83; *p* = 0.001; eta^2^ = 0.81) (Figure 4). The eta-squared value of 0.81 indicates that 81% of the variation in Pb concentration ranks can be explained by the type of processing applied, demonstrating a very strong effect of this variable on the element’s content. Pb concentrations ranged from 0.4447 to 0.4841 μg/mL in dried preparations, 0.3956 to 0.4261 μg/mL in water-based preparations, 0.4566 to 0.4590 μg/mL in fermented preparations, and 0.7914 to 0.7961 μg/mL in tincture preparations. Pairwise comparisons showed that the Pb concentration in aqueous preparations was lower compared to that in the fermented (*p* = 0.03), dried (*p* = 0.006), and alcoholic (*p* < 0.001) forms. In contrast to Cd, the highest Pb levels were observed in tincture, indicating that alcoholic extraction enhanced Pb solubility, while aqueous preparations retained the lowest Pb content.

Overall, these results demonstrate that processing technique had a decisive impact on the concentrations of Cd and Pb in Amanita muscaria preparations: drying favored Cd retention, alcoholic extraction resulted in elevated Pb content, whereas water-based and fermented methods yielded generally lower concentrations of both elements.

## 4. Discussion

### 4.1. Background and Rationale for the Study

In light of the growing trend in the use of intoxicating substances, changes have been made in legislation regarding the possession and sale of new psychoactive substances. As a result, this has led to a slight decrease in the availability of these products. Unfortunately, in response to legislative changes, people began to seek alternatives to NPSs. This gave rise to a trend involving the fly agaric mushroom, which is characterized by psychedelic properties. The controversies surrounding the consumption of the fly agaric mainly relate to poisonings caused by the active compounds found in this species, muscimol and ibotenic acid, which induce stimulation but also central nervous system depression [3,17]. It is worth noting, however, that another risk associated with the consumption of this mushroom is the potential for poisoning by toxic elements. Mushrooms, including the fly agaric, are known for their ability to accumulate metals, including cadmium and lead. Consequently, the use of preparations made from this species may pose a threat to health and even life [3,18,19]. To date, the scientific literature includes only studies on the concentration of toxic elements in dried fly agaric mushrooms. However, there is a lack of research on the levels of cadmium and lead in various popular fly agaric preparations. In light of the growing trend of using preparations from this species, such studies were undertaken. An analysis of cadmium and lead concentrations was carried out in seven fly agaric preparations, including two forms of the tincture: after mineralization and directly.

### 4.2. Cadmium Levels in Amanita muscaria Preparations

The study showed that the concentration of cadmium in dried preparations was significantly higher than those in the alcoholic, aqueous, and fermented forms. In the present study, dried preparations reached 0.1789–0.1888 μg/mL, while water-based preparations contained only 0.0077–0.0314 μg/mL, and fermented samples were as low as 0.0084–0.0097 μg/mL. Tinctures showed intermediate values of 0.0835–0.0854 μg/mL. This demonstrates that drying concentrates Cd strongly, whereas aqueous and fermented processing markedly reduce its levels, with fermentation giving the most pronounced reduction. These differences were statistically significant, with aqueous and fermented samples containing significantly less Cd than dried mushrooms. The highest cadmium content in the dried preparations may have been due to the loss of water during the drying process, which results in the concentration of metals in the dry matter of the mushroom [18,19]. From a practical perspective, the very low Cd concentrations observed in aqueous and fermented preparations suggest that these forms substantially lower potential Cd exposure for consumers of processed *Amanita muscaria*. In the study by Ordak et al. [10], which analyzed differences in heavy metal content in *Amanita muscaria* depending on the harvest season, mushrooms collected in October 2023, the same period during which the samples for the present study were gathered, contained an average of 10.07 µg/g dry weight of cadmium in mushrooms with skin and 7.75 µg/g dry weight in those without skin. The differences in the obtained data may result both from the different collection region and from the fact that whole mushrooms were used for analysis in that study [10], whereas in the present study, only the caps were analyzed. In a study by Chiocchetti et al. [18], the effect of cooking 11 species of dried mushrooms on the content of toxic elements was analyzed. The analysis showed that in 60% of all mushrooms, a reduction in cadmium levels was observed, averaging 48%. The percentage of metals present in the water in which the mushrooms were cooked was also examined in relation to the content in the sample. For cadmium, the result ranged from 11% to 38%. It is also worth noting that the highest level of cadmium was recorded in *Amanita caesarea*, which belongs to the same family as *Amanita muscaria* [18]. The results of the above-mentioned studies are consistent with those obtained in the present research. Cadmium content in cooked *Amanita muscaria* was also lower than that in dried preparations, with a significant reduction of up to 80%. This is due to the solubilization of elements during cooking, allowing part of the cadmium to leach into the water. Another study analyzed the effects of blanching and pickling *Cantharellus cibarius* on cadmium levels. Blanching dried mushrooms resulted in a cadmium reduction ranging from 11 ± 7% to 36 ± 7%. Additionally, pickling the blanched mushrooms further reduced cadmium levels by 42–71%. Comparable observations on the influence of fermentation on Cd mobility have also been reported in other food matrices. For example, a study on cacao bean fermentation demonstrated that Cd concentrations in nibs decreased significantly during the fermentation process, particularly when associated with a reduction in pH, which enhanced Cd mobility [19]. These findings provide further support for the role of fermentation in altering trace element content, consistent with the reduced Cd concentrations observed in fermented Amanita muscaria in the present study. Drewnowska et al. [20] suggested that pickling previously blanched mushrooms can lower cadmium content by as much as 72–91%. Moreover, pickling was shown to be more effective than blanching alone, likely due to better solubility of cadmium complexes in acidic solutions compared to water. In a study conducted by Žitkevičius et al. [21], the levels of cadmium and lead were analyzed in the fruits of *Crataegus monogyna* Jacq., the herb of *Echinacea purpurea* Moench., and the herb of *Leonurus cardiaca* L., as well as their degree of migration into medicinal products such as liquid extracts and tinctures. It was shown that 49.63% of cadmium transfers from the plant material into extracts and tinctures. In another study, the concentration of heavy metals in 25 medicinal plants was analyzed [22]. The results suggest that ethanol-based solutions have stronger extraction properties compared to water. The enhanced migration of cadmium into alcoholic solutions may be due to the breakdown of complexes formed between this element and the plant’s organic matrix.

### 4.3. Lead Levels in Amanita muscaria Preparations

In the case of lead concentration results, statistical significance was also observed, indicating considerable variation in the levels of this element among different forms of *Amanita muscaria* preparations. The highest lead content was found in the tinctures, both those measured directly and after mineralization. The remaining preparations showed similar lead concentration values. In the present study, dried preparations contained 0.4447–0.4841 μg/mL of Pb, water-based forms were lower at 0.3956–0.4261 μg/mL, while fermented samples reached 0.4566–0.4590 μg/mL. The highest levels were observed in tinctures, with 0.7914–0.7961 μg/mL. This pattern indicates that alcoholic extraction markedly increases Pb release, whereas water-based preparations retain the lowest Pb concentrations. These differences were statistically significant, with aqueous samples showing significantly lower Pb levels compared to fermented, dried, and alcoholic forms. From a practical perspective, this highlights that aqueous processing minimizes Pb exposure, while tinctures pose the greatest toxicological risk. In the above-mentioned study reported by Adie et al. [22], the content of toxic elements was analyzed in samples of 25 medicinal plants. The results showed that ethanol has a significantly greater ability to extract lead compared to water. In the case of aqueous extracts, lead was detected in only one out of four samples, whereas in ethanol extracts, it was detected in three out of four tested samples. These findings suggest that ethanol more effectively breaks down complexes formed between the heavy metal and the organic matrix of the raw material, resulting in the release of the element into the extract. In another study, the content of toxic elements in *Crataegus monogyna* Jacq., *Echinacea purpurea* Moench., and *Leonurus cardiaca* L., as well as the medicinal products made from them, was analyzed [21]. The results showed that lead has a higher absorption capacity than cadmium, yet only 10.73% of its amount from the raw material transferred into the extracts and tincture. Apart from the tincture, the highest level of lead was found in dried *Amanita muscaria*. The average content of this toxic element in the dried mushroom with skin was 23.07 µg/g, and without skin 20.03 µg/g. The obtained data may indicate that part of the lead also accumulates in the skin. The high content of this metal in dried preparations may result from the mushroom processing itself. During drying, water is lost, leading to a concentration of the element in the mushroom [18,20]. From a practical perspective, the observation that aqueous preparations consistently contained the lowest Pb concentrations suggests that water-based forms of Amanita muscaria may substantially reduce Pb exposure compared to dried or alcoholic products. Conversely, tinctures represent the most concerning source of Pb, reinforcing the importance of preparation type when assessing toxicological risks. In a study by Kiczorowski et al. [23], the effect of lactic acid fermentation on the chemical composition of *Brassica oleracea* var. *italica*, *Daucus carota*, *Cucumis*, *Capsicum annuum*, and *Beta vulgaris* subsp. *vulgaris* was analyzed. The study covered changes in the content of mineral compounds, heavy metals, vitamins, phenolic compounds, and the overall chemical profile of these plants after fermentation. The analysis showed that the fermentation process led to a decrease in lead content in the tested plants. Furthermore, in fermented *Daucus carota*, *Cucumis*, and *Beta vulgaris* subsp. *vulgaris*, the level of this element was reduced by as much as 18.5% compared to that in the raw versions. The reduction in heavy metal content in fermented plants and mushrooms may be due to the adsorption of metals by the cell walls of fermentative bacteria [23]. These findings are consistent with the moderate Pb levels observed in fermented *Amanita muscaria* samples in the present study, suggesting that fermentation may play a role in partially reducing Pb content. Aqueous preparations of *Amanita muscaria* showed the lowest lead concentrations among all samples. In the study by Chiocchetti et al. [18], cooking 11 mushroom species reduced lead content by an average of 25% in 60% of cases, with 4–12% of the lead migrating into the water. The highest lead level, as with cadmium, was observed in *Amanita caesarea*. The reduction in lead in cooked *Amanita muscaria* likely results from solubilization during cooking, while the low lead content in the infusion and decoction may be due to the formation of stable complexes with the mushroom matrix, limiting its transfer to water.

### 4.4. Toxicological Risks and Health Implications

Considering the above results, it was demonstrated that Cd and Pb accumulate in *Amanita muscaria*, and various processing methods reduce their levels. However, the highest concentrations were found in dried and tincture forms, which are among the most frequently consumed and promoted online. In contrast, aqueous and fermented preparations showed significantly lower levels of these metals. Despite this, even reduced concentrations may pose a threat with repeated or high-dose use. It is important to note that chronic consumption of mushrooms containing heavy metals may significantly affect consumer health. Since all tested forms were analyzed in their final liquid state, reflecting the typical mode of ingestion, the results provide a realistic assessment of actual exposure. This approach enhances the relevance of the findings for evaluating toxicological risks associated with consumption. It also eliminates variability linked to matrix differences, ensuring methodological consistency. As such, the conclusions drawn here can be directly applied to real-world usage scenarios. To place these findings in a regulatory context, maximum residue levels (MRLs) for Cd and Pb in foodstuffs, including mushrooms, are established under food safety legislation. Although specific thresholds for processed products such as tinctures or fermented extracts are not defined, reference to these limits is important for assessing potential consumer risk. Studies on mice have shown that prolonged exposure to Cd leads to metabolic disorders, as well as structural and functional changes in the liver and gut microbiota [24]. In studies on rats, administration of low doses of Cd caused damage to the renal tubules and fibrosis of the glomeruli [25]. Another study indicated that Cd may also contribute to the development of musculoskeletal disorders through oxidative stress, bone metabolism disturbances, and the exacerbation of inflammatory conditions [26]. Cd also exhibits neurotoxic effects. Haider et al. [27] demonstrated that Cd exposure in rats led to increased depressive and anxiety-like behaviors, memory problems, and altered brain enzyme activity. Like Cd, Pb is a toxic element that poses a threat to human health. Research on rats has shown that long-term Pb exposure can cause locomotor disturbances and cerebellar cell damage due to changes in cell signaling and apoptosis [28]. Other studies have suggested that increased Pb exposure alters serotonin levels, resulting in anxiety in rats [29]. Pb has also been found to negatively affect the cardiovascular system. Chronic exposure to low doses of this element led to the development of hypertension and impaired autonomic regulation of cardiovascular function [30]. Findings by Álvarez-Lloret et al. [31] suggest that long-term Pb exposure may cause mineral and microstructural changes in bones, thereby disrupting bone remodeling and increasing the risk of osteoporosis. Thus, comparing the measured Cd and Pb concentrations in Amanita muscaria preparations with existing MRLs provides a clearer toxicological risk assessment. The results indicate that aqueous and fermented preparations present the lowest risk, while tinctures pose the highest potential concern. The measured concentrations of Cd and Pb in various *Amanita muscaria* preparations are cause for concern given the known toxicological profiles of these elements. Although toxic effects were not assessed directly, reference to existing in vivo studies illustrates the potential health risks associated with prolonged exposure. This underlines the importance of monitoring heavy metal content in such products. In this study, the analysis was limited to Cd and Pb due to methodological constraints and their consistent detection at elevated levels in Amanita muscaria. Nevertheless, future research will also include mercury (Hg), arsenic (As), and chromium (Cr), as these elements are known to occur in forest soils and have been reported in other mushroom species. Their quantification in A. muscaria will allow for a more comprehensive assessment of the spectrum of toxic trace elements, seasonal and geographical variation in their accumulation, and the potential health risks associated with their consumption.

## 5. Conclusions

The conducted study demonstrated that *Amanita muscaria* preparations differ significantly in their cadmium and lead content, depending on the method of processing. Dried and alcoholic forms were characterized by the highest concentrations, likely due to water loss during drying and the strong extraction capacity of ethanol. In contrast, aqueous and fermented preparations showed notably lower levels of both toxic elements, which may be explained by the limited solubility of metals in water and by the adsorption or transformation of heavy metals during fermentation. Aqueous and fermented forms showed notably lower concentrations of both toxic elements compared to dried and alcoholic preparations. These findings indicate that while certain preparation techniques may reduce heavy metal exposure, none of the tested forms can be considered entirely safe for consumption. From a toxicological perspective, aqueous and fermented products appear to reduce potential exposure to Cd and Pb; however, the overall toxicological profile of Amanita muscaria preparations remains complex due to the presence of other bioactive and potentially harmful compounds. Importantly, this study is limited to the determination of total Cd and Pb concentrations, without assessing their bioaccessibility or bioavailability. Future research should address consumer exposure by evaluating the fraction of metals released during digestion, which would provide a more direct toxicological relevance.

## Figures and Tables

**Figure 1 toxics-13-00734-f001:**
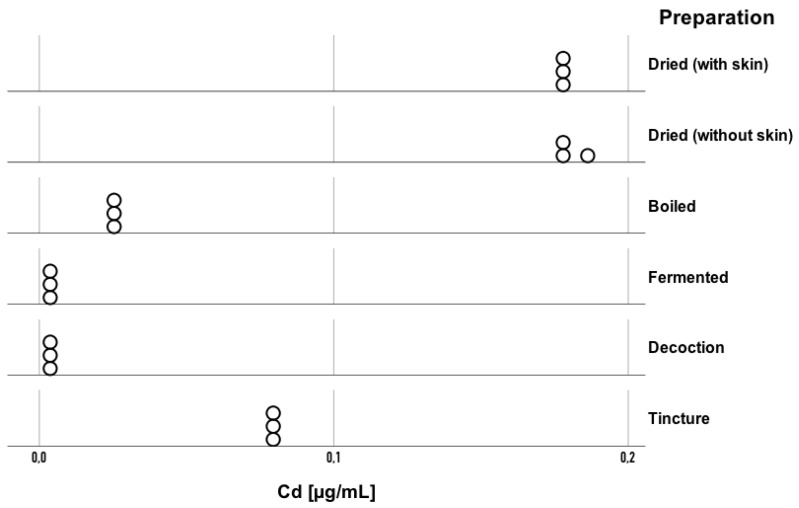
Cd concentrations in the analyzed *Amanita muscaria* preparations. Each point represents an individual replicate measurement (n = 3 per sample).

**Figure 2 toxics-13-00734-f002:**
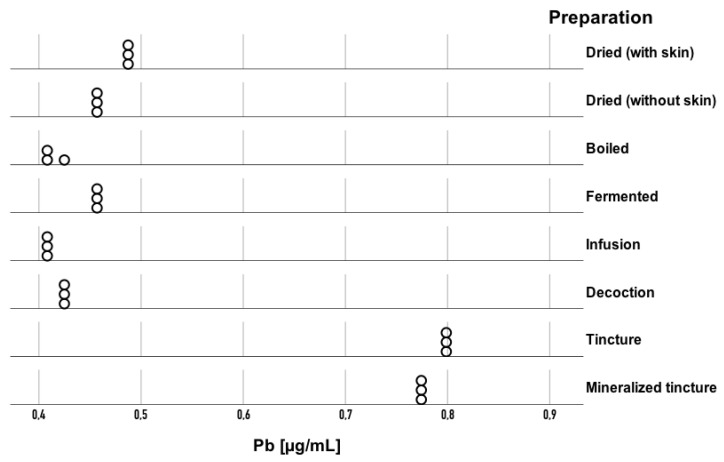
Pb concentrations in the analyzed *Amanita muscaria* preparations. Each point represents an individual replicate measurement (n = 3 per sample).

**Figure 3 toxics-13-00734-f003:**
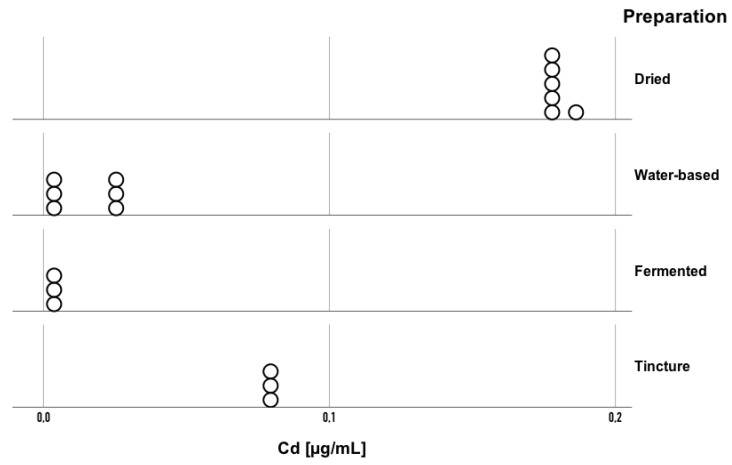
Cd concentrations in Amanita muscaria preparations grouped by processing technique (dried, water-based, fermented, tincture). Each point represents an individual replicate measurement (n = 3 per sample).

**Figure 4 toxics-13-00734-f004:**
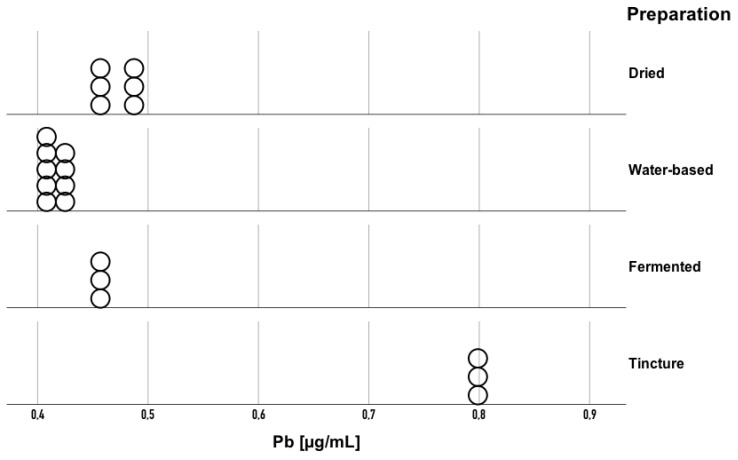
Pb concentrations in Amanita muscaria preparations grouped by processing technique (dried, water-based, fermented, tincture). Each point represents an individual replicate measurement (n = 3 per sample).

## Data Availability

The data that support the findings of this study are available from the corresponding authors upon reasonable request.

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
