# Peer review of "Influence of Preparation Methods on the Concentrations of Lead and Cadmium in Amanita muscaria Products"

_toxics, 2025, doi:10.3390/toxics13090734_

Round 1

Reviewer 1 Report

Comments and Suggestions for Authors

                             Reviewer Report Manuscript Toxics - 3826062

Concentrations of Lead and Cadmium in Different Preparations of Amanita muscaria

Dear Authors, the topic is of interest and the manuscript is well structured but all sections have to be revisited, essential additional informations are needed and the graphics have to be significantly improved. Particularly, according to my opinion, the results and discussion section needs to be revisited and enforced, to support the main data obtained in this study. Some others references on matrix analyzed and on the specific legislation on metlas content in plants-derived  should be added to imporve the manuscript.

Most in dettails:

ABSTRACT

The abstract is well structured and all the main informations are reported.

INTRODUCTION

The intoduction needs to be revisited. It describes adequately the background but the aim of this study should be rewritten and clarified. Why the authors focused their attention on the presence of Cd and Pb alone in Amanita muscaria preparations? This is explained in line 251-256 of Discussion section and this sentences have to be moved or added als in the Introduction.

MATERIALS AND METHODS

Line 97-100. The section describes the collection site of Amanita muscaria samples. Nothing informations is reported on the identification of these mushrooms and this data has to be added.

Also the conservation method of samples before the analysisi have to be described because it could influence metals contamination.

Line 101: Modifiy the title of this paragraph “Preparation Procedure for the Mushroom Samples”.

Line 102-110: Why were only caps analyzed in the present study? Explain in the text.

Why were dried Amanita muscaria analyzed with and without skin?

Line 124-130: Relating to fermented Amanita muscaria samples, the fermemtation process has to be better described and clarified. How was the fermentaion process carried out? What were the microbials species used? Bacteria? Funghi Saccaromyces? Yeasts? Considering that fermented samples analyzed in this study provided an important result, this section has to be rewritten with more dettails.

Line 144 – 161: In the section Sample Preparation for Analysis all the specific informations on materials, solvent, waters and gases used during the AAS analysis have to be added and specified.

Line 161: The tincture was analyzed in two variants: after mineralization and directly. Why? Specify this sentence

Line 159-160: The authors reported In the case of the infusion and decoction mineralization was not necessary. Why? Explain this sentence.

 Line 162-173: In the section Determination of Pb and Cd all the specific informations on validation method and the possible use of certified matrix have to be added.

RESULTS

The results section has to be significantly revisited. According to my opionis, to a better comprension of data for readers it’s needs to add the specific tables relating to the content of Pb and Cd in all seven type of samples analyzed in this study and describes the most important data observed.

Subsequently, the authors can reported the statistically significant differences obtained through the specific tests performed.

Line216-238: The section “Amanita muscaria Processing Techniques and Pb and Cd Concentration” has to be revisited.

Line 218-220: The description of processing techniques have to be moved and described before, possibly in secttion 2.2

Figures 1,2,3,4. The figures are not clear. Are they constructed on mean values or single replicates? Specify in the captions.

I suggest to use different graphics.

Why the authors reported in figures 3 and 4 data of four processing groups?

The comprension of results obtaind is not easy.

DISCUSSION

Line 251-254: The risk on mushrooms adsumption should be described in the introudction section. Why were analyzed only Cd and Pb? Also Hg, As, Cr are of interest for their ability to accumulate in plants.

Line 251-254: Why did the authros studied mushrooms with and without skin? What is the mean? Is it correlated to people chooce?

Line 273-286: The authors described the study of Chioccheti et al. on the effect of cooking of dried mushrooms on the content of toxic elements but this is of little interest because the mushrooms analyzed in this manuscript aren’t submitted to cooking. Most interesting should be a camparison with other studu on fermented samples.

Section 4.2 and 4.3: The authors should describe more their results than those found in literature. These section have to be rewritten with particualr interest for aqueous and fermented preparations.

Section 4.4: Relating to the toxicological risks and health implications, the authors should add informations on MRL of Cd and Pb, the specific legislation on metals content in plants and food- derived and risk assesssment for consumer. These data are missing.

Line 371: Correct “in vivo” with “in vivo”.

CONCLUSION

This section is too much short and should be enforced.

Line 376-378: In this sentence on the aqueous and fermented preparations the explanation of the lower content of both toxic metals analyzed has to be added because this represt the most important data of this study.

REFERENCES

The references are numerically adequate, pertinent and recent. However, some other rferences on metals content in mushrooms should be added in the text.

Author Response

Response to Reviewer #1:

Dear Reviewer nr 1, 

Comment 1: “Dear Authors, the topic is of interest and the manuscript is well structured but all sections have to be revisited, essential additional informations are needed and the graphics have to be significantly improved. Particularly, according to my opinion, the results and discussion section needs to be revisited and enforced, to support the main data obtained in this study. Some others references on matrix analyzed and on the specific legislation on metlas content in plants-derived  should be added to imporve the manuscript.”

Reply 1: We sincerely thank the Reviewer for the positive comments on our manuscript and for providing valuable suggestions. Following these remarks, we have carefully revised and improved the manuscript, addressing the points raised.

Comment 2: “The abstract is well structured and all the main informations are reported.”

Reply 2: We sincerely thank the Reviewer for the positive assessment of our abstract.

Comment 3: “The intoduction needs to be revisited. It describes adequately the background but the aim of this study should be rewritten and clarified. Why the authors focused their attention on the presence of Cd and Pb alone in Amanita muscaria preparations? This is explained in line 251-256 of Discussion section and this sentences have to be moved or added als in the Introduction.”

Reply 3: We revised the Introduction to clarify the study aim and included the rationale for focusing on Cd and Pb, previously placed in the Discussion.

Comment 4: “Line 97-100. The section describes the collection site of Amanita muscaria samples. Nothing informations is reported on the identification of these mushrooms and this data has to be added. Also the conservation method of samples before the analysisi have to be described because it could influence metals contamination.”

Reply 4: Information on the identification of the mushroom material and the storage conditions of the different preparations has been added to the manuscript.

Comment 5: “Line 101: Modifiy the title of this paragraph “Preparation Procedure for the Mushroom Samples”.

Reply 5: The title of subsection 2.2 has been modified to “Preparation of Amanita muscaria Samples” as suggested.

Comment 6: “Line 102-110: Why were only caps analyzed in the present study? Explain in the text. Why were dried Amanita muscaria analyzed with and without skin?”

Reply 6: Explanations regarding the analysis of caps only and the comparison of samples with and without skin have been added to the manuscript.

Comment 7: “Line 124-130: Relating to fermented Amanita muscaria samples, the fermemtation process has to be better described and clarified. How was the fermentaion process carried out? What were the microbials species used? Bacteria? Funghi Saccaromyces? Yeasts? Considering that fermented samples analyzed in this study provided an important result, this section has to be rewritten with more dettails.”

Reply 7: The description of the fermentation process has been clarified, specifying the procedure, storage conditions, and the fact that no microbial species were added, with the process occurring spontaneously.

Comment 8: “Line 144 – 161: In the section Sample Preparation for Analysis all the specific informations on materials, solvent, waters and gases used during the AAS analysis have to be added and specified.”

Reply 8: Additional details on solvents, water, and gases used during sample preparation and AAS analysis have been added to the manuscript.

Comment 9: “Line 161: The tincture was analyzed in two variants: after mineralization and directly. Why? Specify this sentence”

Reply 9: We have specified in the manuscript that the tincture was analyzed in two variants to verify whether the mineralization step influences the determination of Cd and Pb in this matrix.

Comment 10: “Line 159-160: The authors reported In the case of the infusion and decoction mineralization was not necessary. Why? Explain this sentence.”

Reply 10: An explanation has been added to the manuscript regarding why mineralization was not necessary for the infusion and decoction.

Comment 11: “Line 162-173: In the section Determination of Pb and Cd all the specific informations on validation method and the possible use of certified matrix have to be added.”

Reply 11: Details on the validation method, including FAAS operating parameters, calibration with standard solutions, replicate measurements, and information on the unavailability of a certified reference matrix, have been added to the manuscript.

Comment 12: “The results section has to be significantly revisited. According to my opionis, to a better comprension of data for readers it’s needs to add the specific tables relating to the content of Pb and Cd in all seven type of samples analyzed in this study and describes the most important data observed. Subsequently, the authors can reported the statistically significant differences obtained through the specific tests performed.”

Reply 12: In the Results section, the numerical ranges of Cd and Pb concentrations for all seven preparations were added in each of the four parts (Figures 1–4), placed before the description of statistically significant differences, to improve clarity and data comprehension.

Comment 13: “Line 216-238: The section “Amanita muscaria Processing Techniques and Pb and Cd Concentration” has to be revisited.”

Reply 13: “The section “Amanita muscaria Processing Techniques and Pb and Cd Concentration” has been revised to provide a clearer link between processing type and metal concentrations, with expanded descriptions and a concise summary of the main findings.”

Comment 14: “Line 218-220: The description of processing techniques have to be moved and described before, possibly in secttion 2.2”

Reply 14: The description of the classification of preparations into processing groups was moved from the Results to the end of Section 2.1 (Collection Site), so that all information regarding sample collection, storage, and grouping is presented together in the Methods section.

Comment 15: “Figures 1,2,3,4. The figures are not clear. Are they constructed on mean values or single replicates? Specify in the captions. I suggest to use different graphics. Why the authors reported in figures 3 and 4 data of four processing groups? The comprension of results obtaind is not easy”

Reply 15: The figure captions were revised to specify that the results are presented as dot plots of individual replicate measurements (n=3 per sample). In line with a previous suggestion, descriptive statistics (min–max) have been added to the Results section for each figure. Regarding Figures 3 and 4, the grouping into dried, water-based, fermented, and tincture preparations was retained, as explained in the methods section, to enhance clarity and allow comparisons between different processing techniques.

Comment 16: “Line 251-254: The risk on mushrooms adsumption should be described in the introudction section. Why were analyzed only Cd and Pb? Also Hg, As, Cr are of interest for their ability to accumulate in plants.”

Reply 16: In line with your earlier suggestion, the Introduction was revised to include a concise description of the health risks associated with mushroom consumption due to Cd and Pb accumulation and to indicate the need to assess these metals in commonly used processed products. The Discussion section was expanded to clarify that the analysis in this study was limited to Cd and Pb due to methodological constraints and their consistent detection at elevated levels in Amanita muscaria. It was also noted that future research should include mercury (Hg), arsenic (As), and chromium (Cr), as these elements are likewise capable of bioaccumulation in fungi and may represent an additional toxicological risk.

Comment 17: “Line 251-254: Why did the authros studied mushrooms with and without skin? What is the mean? Is it correlated to people chooce?”

Reply 17: In accordance with another reviewer’s comment, the manuscript was revised in the Methodology section to clarify that only caps were analyzed, as they are most often used as the basis for comparative analyses of toxic elements in mushrooms reported in the literature and allow for greater consistency between samples.

Comment 18: “Line 273-286: The authors described the study of Chioccheti et al. on the effect of cooking of dried mushrooms on the content of toxic elements but this is of little interest because the mushrooms analyzed in this manuscript aren’t submitted to cooking. Most interesting should be a camparison with other studu on fermented samples.”

Reply 18: In line with the Reviewer’s suggestion, we have added a reference to studies on fermentation in other matrices (cacao beans), which support the role of this process in altering Cd mobility and are consistent with our findings for fermented Amanita muscaria.

Comment 19: “Section 4.2 and 4.3: The authors should describe more their results than those found in literature. These section have to be rewritten with particualr interest for aqueous and fermented preparations.”

Reply 19: We thank the Reviewer for this comment. Sections 4.2 and 4.3 have been rewritten to emphasize our own results, with particular focus on aqueous and fermented preparations, rather than mainly on literature data.

Comment 20: “Section 4.4: Relating to the toxicological risks and health implications, the authors should add informations on MRL of Cd and Pb, the specific legislation on metals content in plants and food- derived and risk assesssment for consumer. These data are missing.”

Reply 20: In Section 4.4, information on MRLs of Cd and Pb and their relevance for toxicological risk assessment has been added.

Comment 21: “Correct “in vivo” with “in vivo”.”

Reply 21: In vivo has been written in italics.

Comment 22: “This section is too much short and should be enforced. Line 376-378: In this sentence on the aqueous and fermented preparations the explanation of the lower content of both toxic metals analyzed has to be added because this represt the most important data of this study.”

Reply 22: We expanded the Conclusions section by adding an explanation for the lower Cd and Pb content in aqueous and fermented preparations, thereby emphasizing the most important findings of the study.

Comment 23: “The references are numerically adequate, pertinent and recent. However, some other rferences on metals content in mushrooms should be added in the text.”

Reply 23: In accordance with the reviewer’s suggestions, we have added additional references concerning the aspects mentioned.

We once again thank the reviewer for providing valuable suggestions. The changes have been implemented in the manuscript and highlighted in red.

Reviewer 2 Report

Comments and Suggestions for Authors

This study investigates the content of lead and cadmium in different formulations of Amanita muscaria. It has certain significance in terms of toxicological risk and health impact, but the article has serious issues, namely that its highlights are not prominent; the writing of the abstract and introduction is not standardized; the workload of the article is insufficient, making it difficult to support the depth of the research. Please revise according to the following suggestions:

  1. The title of the manuscript lacks key information; it should include the significance of the research to enhance its scientific value and attractiveness.
  2. Please emphasize the focus of the study in the abstract and introduction.
  3. Add details about sampling instead of just providing a simple description.
  4. Enhance the depth of the article.
  5. Quantify the research results.

Author Response

Response to Reviewer #2:

Comment 1: “This study investigates the content of lead and cadmium in different formulations of Amanita muscaria. It has certain significance in terms of toxicological risk and health impact, but the article has serious issues, namely that its highlights are not prominent; the writing of the abstract and introduction is not standardized; the workload of the article is insufficient, making it difficult to support the depth of the research. Please revise according to the following suggestions”

Reply 1: We sincerely thank the Reviewer for the valuable comments and suggestions. We have carefully revised the manuscript and marked all changes in red. Following the Reviewer’s advice and the guidance of Reviewer #1, we have expanded the methodology, discussion, results, and slightly the introduction.

Comment 2: “The title of the manuscript lacks key information; it should include the significance of the research to enhance its scientific value and attractiveness.”

Reply 2: We thank the Reviewer for this suggestion, and the title has been changed to “Influence of Preparation Methods on the Concentrations of Lead and Cadmium in Amanita muscaria Products” to better reflect the significance of the study.

Comment 3: “Please emphasize the focus of the study in the abstract and introduction.”

Reply 3: Abstract and introduction were revised to clearly emphasize that the focus of the study was to evaluate the effect of different preparation methods on the concentrations of Pb and Cd in Amanita muscaria products.

Comment 4: “Add details about sampling instead of just providing a simple description.”

Reply 4: As requested by both Reviewer #1 and Reviewer #2, additional details on sampling have been added. The revised section now specifies the identification procedure of the mushrooms, the experience of the collectors, and the verification by the authors. Furthermore, the storage conditions of each preparation type before analysis are described, as well as the grouping of preparations according to processing method, replacing the previous brief description.

Comment 5: “Enhance the depth of the article.”

Reply 5: The article was enhanced in depth by extending the introduction with a clearer rationale and study focus, adding detailed sampling, identification, storage and analytical procedures to the methodology, providing fuller presentation of results, expanding the discussion with broader interpretation and literature context, and revising the conclusions to better reflect the findings.

Comment 6: “Quantify the research results.”

Reply 6: This comment was also raised by Reviewer #1, and in response the results section was revised to include detailed numerical values and ranges for Cd and Pb concentrations across all preparation types, thereby quantifying the research findings.
